# Identifying genetic variants associated with ritodrine-induced pulmonary edema

Seung Mi Lee[1‡], Yoomi Park[2‡], Young Ju Kim[3], Han-Sung Hwang[4], Heewon Seo[2,5], Byung-Joo Min[2], Kye Hwa Lee[2¤], So Yeon Kim[1], Young Mi Jung[1], Suehyun Lee[6], Chan-Wook Park[1], Ju Han Kim[2‡]*, Joong Shin Park[1‡]*

1 Department of Obstetrics and Gynecology, Seoul National University College of Medicine, Seoul, Korea,
2 Division of Biomedical Informatics, Seoul National University College of Medicine, Seoul, Korea,
3 Department of Obstetrics and Gynecology, Ewha Womans University College of Medicine, Seoul, Korea,
4 Department of Obstetrics and Gynecology, Konkuk University School of Medicine, Seoul, Korea,
5 Princess Margaret Cancer Centre, University Health Network, Toronto, ON, Canada, 6 Department of Biomedical Informatics, Konyang University, Daejeon, Korea

¤ Current address: Department of Information Medicine, Asan Medical Center, University of Ulsan College of Medicine, Seoul, Korea
‡ SML and YP contributed equally to this work as co-first authors. JHK and JSP also contributed equally to this work as co-corresponding authors.
* jsparkmd@snu.ac.kr (JSP); juhan@snu.ac.kr (JHK)

**Data Availability Statement:** The authors confirm that the genetic data supporting the findings of this study are available within the article and its supplementary materials. Clinical data cannot be shared publicly because it contains potentially

## Abstract

### Introduction

Ritodrine is one of the most commonly used tocolytics in preterm labor, acting as a ß2-adrenergic agonist that reduces intracellular calcium levels and prevents myometrial activation. Ritodrine infusion can result in serious maternal complications, and pulmonary edema is a particular concern among these. The cause of pulmonary edema following ritodrine treatment is multifactorial; however, the contributing genetic factors remain poorly understood. This study investigates the genetic variants associated with ritodrine-induced pulmonary edema.

### Methods

In this case-control study, 16 patients who developed pulmonary edema during ritodrine infusion [case], and 16 pregnant women who were treated with ritodrine and did not develop pulmonary edema [control] were included. The control pregnant women were selected after matching for plurality and gestational age at the time of tocolytic use. Maternal blood was collected during admission for tocolytic treatment, and whole exome sequencing was performed with the stored blood samples.

### Results

Gene-wise variant burden (GVB) analysis resulted in a total of 71 candidate genes by comparing the cumulative effects of multiple coding variants for 19729 protein-coding genes between the patients with pulmonary edema and the matched controls. Subsequent data analysis selected only the statistically significant and deleterious variants compatible with

identifying or sensitive patient information. The Institutional Review Boards and Ethics Committee of Seoul National University Hospital (SNUH), Ewha Woman's University Hospital (EUMC), and Konkuk University Hospital (KUH) are imposing the restrictions on the data collected in this study. Please contact the following e-mail or phone number for data access. (SNUH Email: irb@snuh. org, Tel: +82-2-2072-0694; EUMC Email: info@ewhactc.co.kr, Tel: +82-2-2650-5114; KUH Email: 20190508@kuh.ac.kr, Tel: +82-2-2030-6542).

**Funding:** This research was supported by a grant (14172MFDS178) from the Ministry of Food and Drug Safety and the Korean Health Technology R&D Project by the Ministry of Health and Welfare in the Republic of Korea (HI18C2386). The funders played no role in study design, data collection and analysis, decision to publish, or preparation of the manuscript.

**Competing interests:** The authors have declared that no competing interests exist

**Abbreviations:** GVB, Gene-wise variant burden; cAMP, cyclic adenosine monophosphate; PKA, protein kinase A; WES, Whole exome sequencing; CPT2, carnitine palmitoyltransferase 2; ADRA1A, Adrenoceptor Alpha 1A; 1KGP, 1000 Genomes Project, Phase 3; SIFT, Sorting Intolerant From Tolerant; CADD, Combined Annotation Dependent Depletion.

ritodrine-induced pulmonary edema. Two final candidate variants in *CPT2* and *ADRA1A* were confirmed by Sanger sequencing.

## Conclusions

We identified new potential variants in genes that play a role in cyclic adenosine monophosphate (cAMP)/protein kinase A (PKA) regulation, which supports their putative involvement in the predisposition to ritodrine-induced pulmonary edema in pregnant women.

## Introduction

Preterm birth is one of the leading causes of perinatal mortality and/or morbidity, and preterm labor constitutes more than one third of the precursors of preterm birth [1]. In women with preterm labor, tocolytics are commonly used to delay preterm birth for antenatal corticosteroids administration and transfer to referral hospitals.

Ritodrine is one of the FDA-approved tocolytics and has been reported to delay preterm birth in patients with preterm labor [2–4]. However, ritodrine can cause several side effects, and ritodrine-induced pulmonary edema is one of the rarest yet most serious side effect that can lead to death [5, 6]. Despite the severity of pulmonary edema, the pathophysiologic mechanism for its development following ritodrine administration remains unknown.

To date, there has been a paucity of information regarding genetic variants associated with ritodrine-induced pulmonary edema. Whole exome sequencing (WES) is being used to find genetic variants associated with various kinds of human diseases. Using WES, we can identify candidate genes, not only for Mendelian disorders, common diseases, and cancer, but also rare variants for complex diseases [7, 8]. Thus, if we can identify patients who are at risk of developing serious complications, such as pulmonary edema, other tocolytics may be considered, such as calcium channel blockers, oxytocin-receptor antagonists, or prostaglandin inhibitors by WES.

In the current study, we examined the genetic variants associated with ritodrine-induced pulmonary edema. Using WES data, we compared gene-wise variant burden (GVB) [9–11] for each of the 19,729 protein-coding gene between the 16 patients suffering from ritodrine-induced pulmonary edema and their 16 matched controls without symptoms after drug use under similar gestational conditions. *In silico* analysis of deleterious variants was performed, producing two putative East Asian population-specific variants, rs2229291 (*CPT2*) and rs2229126 (*ADRA1A*), located in genes with roles compatible with tocolytic-associated pulmonary edema. Next generation sequencing analysis results for two final candidates were confirmed by Sanger sequencing, with 100% concordance. In summary, we identified two novel potential variants that could help identify patients at high-risk of developing ritodrine-induced pulmonary edema.

## Materials and methods

### Study design

A case-control study was designed to evaluate the genetic variants associated with ritodrine-induced pulmonary edema. Among pregnant women who were admitted for preterm labor between 2000 and 2015 in Seoul National University Hospital, Ewha Woman's University Hospital, or Konkuk University Hospital, the study population consisted of 16 patients who

developed pulmonary edema during ritodrine infusion [case]. As a control group, 16 pregnant women who were treated with ritodrine and did not develop pulmonary edema [control] were included. The control pregnant women were selected based on plurality and gestational age at the time of tocolytics use. This study was approved by the Institutional Review Boards of Seoul National University Hospital, Ewha Woman's University Hospital, and Konkuk University Hospital (IRB Nos. ECT 06–127–7 and KUH1040034).

## Patient and control samples for WES

Maternal blood was collected during admission for preterm labor, and the blood samples were stored at -70˚C until analysis. Exome sequencing was performed with stored maternal blood using the Ion AmpliSeq™ Exome panel (Thermo Fisher, USA) to screen coding sequence regions of the entire genome. This panel included the exome of 19,072 genes, and the size of the total targeted region was 57.7 Mb. The panel contained 293,903 primer pairs that were multiplexed into 12 pools to avoid primer-dimer formation and interference during polymerase chain reaction (PCR). The range of amplicons amplified by these oligo primer pairs were from 125 to 275 bp, and the rate of 'on target' coverage for this panel was 95.69%. PCR assays were performed directly to amplify 100 ng of genomic DNA samples extracted from peripheral blood to collect target regions using oligo primer pairs of the panel. Reaction parameters were as follows: 99˚C for 2 min, followed by 10 cycles of 99˚C for 15 s, 60˚C for 16 min, and 10˚C for 1 min. Following amplification, library construction was performed using the Ion Ampliseq Library Kit Plus (Thermo Scientific, Waltham, MA) as described in the manufacturer's instructions. The libraries were quantified using an Agilent 2100 Bioanalyzer (Agilent, Santa Clara, CA) and subsequently diluted to ~10 pM. Thereafter, 33.3 μL of the barcoded libraries were combined in sets of three barcodes. These combined libraries were sequenced using the Ion Proton platform with PI chip V3 (Thermo Scientific), following the manufacturer's instructions.

## Data analysis

Reads were mapped to the human genome reference sequence (hg19/GRCh37) using Torrent Suite (v4.4). Aligned reads were sorted and indexed via SAMTools (v0.1.19) [12]. Genome Analysis Toolkit (GATK, v2.8.1) [13–15] was used for local realignment, and base recalibration (dbSNP137, and Mills and 1000 Genome Project gold-standard indels from the hg19 sites). Variant calling was performed using HaplotypeCaller in GATK. ANNOVAR (http://annovar.openbioinformatics.org/) [16] was used to annotate variants. To remove likely false positives, only variants previously identified in 1KGP (1000 Genomes Project, Phase 3) [17] were included for further analysis. Using SnpEff 4.1 (build 2015-01-07) [18], functional variants (missense, stop gained, stop lost, start lost) were selected. A gene-wise variant burden (GVB) score was calculated to predict the cumulative genetic effect for all coding variants of a gene. For each individual, the quantitative value for 19,729 protein-coding genes calculated by the geometric mean of SIFT (Sorting Intolerant From Tolerant) [19] annotated variants in a gene was obtained. Average GVB scores between patients and matched controls were compared by using Student's *t*-test. To pinpoint susceptibility loci, Fisher's exact test was conducted for variants in genes that showed statistically significant associations. We used two *in silico* bioinformatics tools to select for deleterious variants that fulfilled one or more criteria with SIFT $\leq$ 0.05 and Combined Annotation Dependent Depletion (CADD) $\geq$ 15 [20]. The association of biological function, drugs and pathways of genes containing the candidate variants were reviewed via KEGG Drug (https://www.genome.jp/kegg/drug/) [21], KEGG Pathway (https://www.genome.jp/kegg/ pathway.html), and Drug2Gene database [22].

## Subsequent validation

For two final candidate loci, we performed independent validation using Sanger sequencing in the 32 individuals. Exon 4 of *CPT2*, including the rs2229291 position, and Exon 2 of *ADRA1A*, including the rs2229126 position, were amplified for Sanger sequencing. PCR assays were performed directly to amplify 20 ng genomic DNA samples to collect target regions using oligo-primer pairs. Reaction parameters were as follows: 95˚C for 5 min, followed by 35 cycles of 95˚C for 30 s, 58˚C for 30 s, 72˚C for 1 min, and 72˚C for 10 min. The RBC HiYield Gel/PCR DNA Mini Kit (Taipai County 220, Taiwan) was used to purify DNA in PCR products. Following purification, PCR samples were directly sequenced using an ABI 3100 semi-automated sequencing analyzer (Applied Biosystems, Lincoln Center Drive Foster City, CA, USA). DNA sequences were analyzed using FinchTV version 1.4.0 (Geospiza, Inc., Seattle, WA, USA).

## Statistical analysis

Variant analysis was conducted using three association tests, including Fisher's exact test under both dominant and recessive models, and a Cochran-Armitage trend test. A comparison of continuous variables between groups was performed using the independent *t*-test or the Mann-Whitney *U*-test. A *p*-value of $< .05$ was considered statistically significant. SPSS version 21.0 software (IBM Inc., Armonk, NY) was used for the analysis.

## Results

### Subject population

Table 1 shows the clinical characteristics of the study population. As a result of matching, the frequency of twin pregnancy and gestational age at the time of tocolytic treatment were not different between both groups. In addition, parity and gestational age at delivery were not different between both groups. Cases with pulmonary edema had a higher maternal age and a higher risk of cesarean delivery; however, this difference did not reach statistical significance ($p = .077$).

### Candidate variant prioritization and validation

To prioritize potential variants associated with ritodrine-induced pulmonary edema, we systematically applied several filtering steps (S1 Fig). Of the 229,813 variants identified in 32 individuals, only variants previously reported in 1KGP were included for further analysis. For

**Table 1. Baseline clinical, anthropometric, biochemical, and metabolic features and pregnancy outcomes of the study population.**

| Characteristics | Control (n = 16) | Pulmonary edema (n = 16) | *P*-value |
|---|---|---|---|
| Maternal age | 31 (27–36) | 34 (25–41) | 0.179 |
| Nulliparity | 13 (81%) | 11 (69%) | 0.685 |
| Twin pregnancy | 5 (31%) | 5 (31%) | 1.000 |
| Diagnosis of admission (PTL) | 11 (69%) | 11 (69%) | 1.000 |
| Gestational age at tocolytics | 30.7 (24.4–33.0) | 30.5 (21.4–34.3) | 0.985 |
| Corticosteroids during tocolytics | 15 (94%) | 14 (88%) | 1.000 |
| Gestational age at diagnosis of PE | | 31.4 (23.7–34.7) | (-) |
| Gestational age at delivery | 32.8 (25.9–40.1) | 32.1 (23.1–41.4) | 0.792 |
| Cesarean delivery | 5 (31%) | 10 (63%) | 0.077 |

Data are presented as the proportion (%) or median (range).

**Table 2. Variants significantly associated with ritodrine-induced pulmonary edema.**

| Gene (rsID) | Group | Allelic Frequency | | | | REF | HET | HOM | Dominant model | | Recessive model | | CATT |
|---|---|---|---|---|---|---|---|---|---|---|---|---|---|
| | | REF | ALT | P | OR (95% CI) | | | | P | OR (95% CI) | P | OR (95% CI) | P |
| *CPT2* (rs2229291) | Pulmonary Edema (+), N = 16 | 17 | 15 | 0.014 | 4.6 (1.3–19.5) | 5 | 7 | 4 | 0.076 | 4.6 (0.9–28.1) | 0.101 | Inf (0.7-Inf) | 0.018 |
| | Matched Control, N = 16 | 27 | 5 | | | 11 | 5 | 0 | | | | | |
| *ADRA1A* (rs2229126) | Pulmonary Edema (+), N = 16 | 27 | 5 | 0.053 | Inf (1.0-Inf) | 11 | 5 | 0 | 0.043 | Inf (1.1-Inf) | 1 | 0 (0-Inf) | 0.080 |
| | Matched Control, N = 16 | 32 | 0 | | | 16 | 0 | 0 | | | | | |

REF, Reference; ALT, Alternative; HET, Heterozygous genotype; HOM, Homozygous genotype; CATT, Cochran–Armitage trend test.

19,729 protein-coding genes, we calculated the GVB score, which represents the aggregated effects of genetic variants in a gene, and 71 genes showed statistically significant differences in the average GVB scores between the pulmonary edema group and normal controls. To identify exact variants with a potentially high functional impact on ritodrine-induced pulmonary edema, 43 variants showing an association signal for Fisher's exact test with a $p < 0.05$ were selected. Among these, 16 nonsynonymous variants predicted as deleterious by meeting at least one out of two *in silico* prediction tools were identified (S1 Table). Candidate variants within genes previously implicated in ritodrine-induced adverse drug reactions or included in functionally interesting pathways were prioritized. Finally, the two final candidate variants located in *CPT2* and *ADRA1A* were selected to be responsible for ritodrine-induced pulmonary edema (Table 2). Of these associations, *CPT2* rs2229291, known as Perhexiline (calcium channel blocker) target gene on KEGG Drug, was specific for East Asian individuals, in whom the G allele is common (17%) with a substantial effect size (OR = 4.6 (1.3–19.5), Fisher's exact test *p*-value = 0.014). *ADRA1A* rs2229126, which has been implicated to have a relationship with ritodrine via the Drug2Gene database, had a marginal association signal (OR = Inf (1.0-Inf), Fisher's exact test *p*-value = 0.053). *CPT2* rs2229291 remained statistically significant even after adjustment for the cesarean delivery effect (multiple linear regression *p*-value = 0.048, S2 Table). Individuals carrying a cumulative number of risk alleles were at increased risk for experiencing ritodrine-induced pulmonary edema compared to the control group with zero or one risk allele (Table 3). The distribution of nonsynonymous variants in *ADRA1A* and *CPT2* genes for 32 individuals is summarized in S3 Table.

## Subsequent validation by Sanger sequencing

The two final candidate variants in *CPT2* and *ADRA1A* were confirmed by Sanger sequencing. Sanger sequences for each of the two loci in 32 individuals were perfectly concordant with NGS sequences (S2–S5 Figs). The results showed that individuals with ritodrine-induced pulmonary edema had a significantly higher number of homozygous (*CPT2* rs2229291, GG) or

**Table 3. Frequency distribution of the number of patients carrying 0 to 4 risk alleles at the two candidate positions.**

| Gene (rsID) | Number of risk alleles | Risk allele carriers (rs2229126 or rs2229291) | | P | OR (95% CI) |
|---|---|---|---|---|---|
| | | Pulmonary Edema (+), N = 16 | Matched Control, N = 16 | | |
| *CPT2* (rs2229291) or *ADRA1A* (rs2229126) | Total (≥ 1) | 13 (81.25) | 5 (25.00) | 0.011 | 8.8 (1.5–71.0) |
| | 0 | 3 (18.75) | 11 (68.75) | 0.011 | 0.1 (0.0–0.7) |
| | 1 | 6 (37.50) | 5 (31.25) | 1 | 1.3 (0.2–7.4) |
| | 2 | 7 (43.75) | 0 (0.00) | 0.007 | Inf (1.9-Inf) |
| | 3 | 0 (0.00) | 0 (0.00) | 1 | 0 (0-Inf) |
| | 4 | 0 (0.00) | 0 (0.00) | 1 | 0 (0-Inf) |

heterozygous (*CPT2* rs2229291, TG) variants than those without. Variants for *ADRA1A* (rs2229126, TA) were only identified in individuals with pulmonary edema in the heterozygous state (Fig 1), which implies that life-threatening toxicity could occur following ritodrine use in homozygous *ADRA1A* variant carriers.

## Discussion

In the current study, we demonstrated that two variants in *CPT2* and *ADRA1A* were selected as candidate variants for pulmonary edema. These new potential variants play a role in cAMP/PKA regulation, which supports their putative involvement in the predisposition to ritodrine-induced pulmonary edema in pregnant women.

In clinical practice, ritodrine is one of the most commonly prescribed medications to inhibit preterm labor, especially in Europe and Asia [2, 23]. Ritodrine is a β-adrenergic receptor agonist that acts by reducing intracellular ionized calcium levels and preventing the activation of myometrial contractile proteins. Ritodrine acts at the β2-adrenergic receptor, which activates adenylyl cyclase, thereby increasing cyclic adenosine monophosphate (cAMP) and activating protein kinase A (PKA) [24]. PKA inactivates the myosin light chain kinase, resulting in muscle relaxation.

Pulmonary edema is one of the most serious side effects of ritodrine. The mechanism of pulmonary edema is multifactorial. The conditions related to preterm labor, such as multiple gestation, antenatal corticosteroids, and intravenous infusion of large amounts of crystalloid, can contribute to the development of pulmonary edema. In addition, ritodrine has been shown to decrease total pulmonary resistance and cause retention of sodium and water, resulting in volume overload [25].

In the current study, we identified two final candidate variants in *CPT2* and *ADRA1A* associated with ritodrine-induced pulmonary edema. *ADRA1A* is a gene encoding the β-adrenergic receptor, which is associated with a calcium channel. The receptor-channel complex contains a G-protein, adenylyl cyclase, and cAMP-dependent kinase, which are related to the mechanism of ritodrine-induced tocolysis. β-adrenergic receptors are also present, not only in the smooth muscle of the uterus but also throughout the lung tissue [26]. Recent reports suggest that β-adrenergic receptors participate in the maintenance of alveolar fluid balance in lung tissue. β2-knockout mice showed decreased ability in the removal of excess water from the airspace and had pulmonary edema following acute lung injury [27]. It is suggested that β-adrenergic receptors participate in the regulation of active Na+ transport for removal of excessive fluid out of the alveolar airspace [28]. In addition, β-adrenergic receptors can enhance endothelial barrier function via inhibition of endothelial contraction and intracellular gap formation, and can regulate surfactant secretion from alveolar cells [29–31]. Thus, decreased function in the β-adrenergic receptor can result in abnormalities in maintenance of alveolar fluid, endothelial barrier function, and surfactant secretion, all of which can contribute to the development of pulmonary edema.

*CPT2* encodes carnitine palmitoyltransferase 2 (CPT2), an enzyme required for oxidation of long-chain fatty acids within mitochondria. The heart is a highly active organ that heavily relies on mitochondrial oxidative metabolism, and if disrupted, can lead to cardiac hypertrophic remodeling and altered cardiac function. The *CPT2*-deficient mouse model developed cardiac hypertrophy and systolic dysfunction, with a significant reduction in blood ejection fraction [32]. In addition, *CPT2*-deficient mice showed increased vascular permeability in the kidney, spleen, and lung, probably due to impaired junctional properties and decreased barrier function in endothelial cells [33]. Overall, the altered dysfunction in *CPT2* may contribute to the pathogenesis of pulmonary edema through decreased function in the right ventricle and

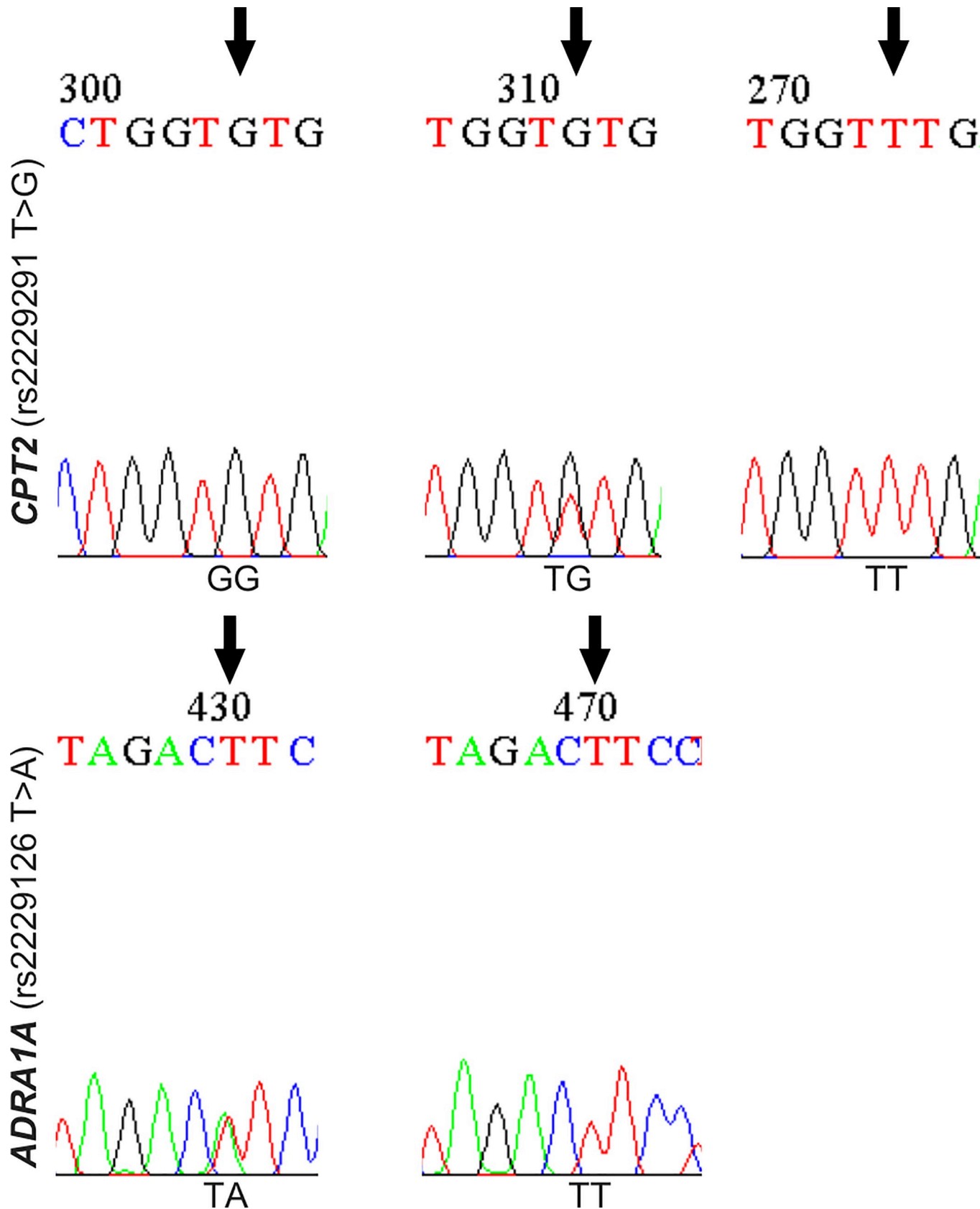

**Fig 1. Results of Sanger sequencing for the representative individuals on which whole exome sequencing was performed.**

increased vascular permeability in lung tissue. There has been a report of decreased activity of *CPT2* with rs2229291 in hyperthermic conditions, resulting in impaired fatty acid oxidation and a mitochondrial energy crisis in vascular endothelial cells [34, 35]; however, the direct functional impact of the variant on ritodrine-induced pulmonary edema should be further evaluated in future studies.

To our knowledge, this is the first study to examine the possible variants associated with ritodrine-induced pulmonary edema using WES. Recently, previous studies discussed possible genetic contributions to the pathogenesis of side effects following ritodrine use; however, these studies evaluated patients not only with pulmonary edema (n = 4 in the study of Seo), but also with nonspecific symptoms including dyspnea, tachycardia, and palpitation [10, 36]. From the database of multiple referral hospitals, we identified 16 patients with pulmonary edema, although its occurrence is one of the rarest side effects. However, the current study only evaluated patients in Korea, and further studies in other ethnicities or races are necessary to validate our results. In addition, the small number of this study population is another limitation of the current study. Future directions should focus on appropriate experimental evaluation of how candidate variants affect the pharmacokinetics of ritodrine and pulmonary edema.

The National Institutes of Health defines precision medicine as the tailoring of medical treatment to the individual characteristics of each patients [37]. Precision medicine implies the ability to classify patients into subpopulations according to their susceptibility to a particular disease/prognosis, or their potential response to a specific treatment. With these strategies, preventive or therapeutic intervention can be concentrated on specific populations who will benefit from these interventions and not experience side effects. Although precision medicine has not been universally introduced in obstetrics, the current study shows that we may identify subpopulations who will experience severe side effects following ritodrine administration by WES.

## Conclusions

In conclusion, we identified new potential variants in genes that play a role in cAMP/PKA regulation, which supports their putative involvement in the predisposition to ritodrine-induced pulmonary edema in pregnant women.

## Supporting information

**S1 Fig. Schematic of the data analysis steps.**
(DOCX)

**S2 Fig. Results of Sanger sequencing of *CPT2* (rs2229291) for the individuals with ritodrine-induced pulmonary edema.**
(DOCX)

**S3 Fig. Results of Sanger sequencing of *CPT2* (rs2229291) for the individuals without ritodrine-induced pulmonary edema.**
(DOCX)

**S4 Fig. Results of Sanger sequencing of *ADRA1A* (rs2229126) for the individuals with ritodrine-induced pulmonary edema.**
(DOCX)

**S5 Fig. Results of Sanger sequencing of *ADRA1A* (rs2229126) for the individuals without ritodrine-induced pulmonary edema.**
(DOCX)

**S1 Table. Frequency distribution for 16 deleterious variants significantly associated with ritodrine induced pulmonary edema.**
(DOCX)

**S2 Table. Variants significantly associated with ritodrine-induced pulmonary edema.**
(DOCX)

**S3 Table. Distribution of coding variants in *ADRA1A* and *CPT2*.**
(DOCX)

## Author Contributions

**Conceptualization:** Seung Mi Lee, Young Ju Kim, Han-Sung Hwang, Ju Han Kim.

**Data curation:** Seung Mi Lee, Yoomi Park, Heewon Seo, Kye Hwa Lee, Young Mi Jung, Suehyun Lee, Chan-Wook Park.

**Formal analysis:** Yoomi Park, Heewon Seo, Byung-Joo Min, Kye Hwa Lee, Young Mi Jung, Suehyun Lee.

**Funding acquisition:** Joong Shin Park.

**Investigation:** Young Ju Kim, Han-Sung Hwang, So Yeon Kim, Young Mi Jung, Chan-Wook Park.

**Methodology:** So Yeon Kim, Young Mi Jung, Suehyun Lee, Chan-Wook Park.

**Project administration:** Young Ju Kim, Han-Sung Hwang, Ju Han Kim, Joong Shin Park.

**Resources:** Byung-Joo Min.

**Software:** Yoomi Park, Heewon Seo.

**Supervision:** Joong Shin Park.

**Writing – original draft:** Seung Mi Lee, Heewon Seo.

**Writing – review & editing:** Joong Shin Park.

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
