## [Decision Letter · Decision Letter 0]

27 Aug 2020

PONE-D-20-22975

Identifying genetic variants associated with ritodrine-induced pulmonary edema

PLOS ONE

Dear Dr. Kim,

Thank you for submitting your manuscript to PLOS ONE. After careful consideration, we feel that it has merit but does not fully meet PLOS ONE’s publication criteria as it currently stands. Therefore, we invite you to submit a revised version of the manuscript that addresses the points raised during the review process.

An expert in the field handled your manuscript, we are very thankful for their time and efforts. Although interest was found in your study, some concerns arose during review that overshadowed this enthusiasm. Notably, the authors need to expand on what their findings about these select genes mean in regards to pulmonary edema and there are questions about the data analysis. Please respond to ALL of the reviewer's comments in your revised manuscript.

We look forward to receiving your revised manuscript.

Kind regards,

Frank T. Spradley

Academic Editor

PLOS ONE

4. Your ethics statement must appear in the Methods section of your manuscript. If your ethics statement is written in any section besides the Methods, please move it to the Methods section and delete it from any other section. Please also ensure that your ethics statement is included in your manuscript, as the ethics section of your online submission will not be published alongside your manuscript.

Reviewers' comments:

Reviewer's Responses to Questions

**Comments to the Author**

1. Is the manuscript technically sound, and do the data support the conclusions?

Reviewer #1: Yes

2. Has the statistical analysis been performed appropriately and rigorously? 

Reviewer #1: Yes

3. Have the authors made all data underlying the findings in their manuscript fully available?

Reviewer #1: Yes

4. Is the manuscript presented in an intelligible fashion and written in standard English?

Reviewer #1: No

5. Review Comments to the Author

Reviewer #1: 1: The author and research team provide another aspect of genetic variants, distinguishing from their previous publication in year 2018 (title: Deleterious genetic variants in ciliopathy genes increase risk of ritodrine-induced cardiac and pulmonary side effects ). In this submitting paper, the authors empathize particular two gene variants (CPT2 & ADRA1A), an interesting question emerged that how ciliapathy related genes or CPT2 & ADRA1A really functions or impact during ritodrine-induced pulmonary edema. Authors could elaborate it or provide more insights within this article.

2: From the results the authors reveal that (CPT2 rs2229291, GG) , (CPT2 rs2229291, TG) and ADRA1A (rs2229126, TA) are significant to ritodrine-induced pulmonary edema. In discussion, authors try to interpret these clinical observations could be causative to the adverse effect in terms of basic studies (such as β2-knockout mouse and CPT2-deficient mouse models). How influential to these genetic variants it will be and any other previous cell-based studies mentioned the function of these genetic variants in different experimental settings.

3: Certain statistics and linguistic could be fortified and precise.

6. PLOS authors have the option to publish the peer review history of their article (what does this mean?). If published, this will include your full peer review and any attached files.

Reviewer #1: **Yes: **Dr. Heng-Liang Lin

---

## [Author Response · Author response to Decision Letter 0]

12 Sep 2020

## Dear Editor:

Thank you very much for the additional comments and suggestions. We have modified the manuscript according to the comments below.

RESPONSE ==> We formatted our manuscript according to the appropriate style manual.

RESPONSE ==> We had uploaded our study’s minimal underlying data set as Supporting Information file (S3 Table).

RESPONSE ==> We included captions for our Supporting Information files at the end of our manuscript.

4. Your ethics statement must appear in the Methods section of your manuscript. If your ethics statement is written in any section besides the Methods, please move it to the Methods section and delete it from any other section. Please also ensure that your ethics statement is included in your manuscript, as the ethics section of your online submission will not be published alongside your manuscript.

RESPONSE ==> We added ethics statement in the Methods section.

We hope this is sufficient for the manuscript to be accepted for publication in PLOS ONE.

On behalf of all the co-authors

Yours sincerely 

Ju Han Kim

##Dear Reviewer:

We greatly appreciate the valuable comments, which have helped to improve the quality of our manuscript. We are hereby submitting a version of the manuscript that has been revised in accordance with your comments. Our point-by-point responses to your comments appear below.

Q1. The author and research team provide another aspect of genetic variants, distinguishing from their previous publication in year 2018 (title: Deleterious genetic variants in ciliopathy genes increase risk of ritodrine-induced cardiac and pulmonary side effects). In this submitting paper, the authors empathize particular two gene variants (CPT2 & ADRA1A), an interesting question emerged that how ciliapathy related genes or CPT2 & ADRA1A really functions or impact during ritodrine-induced pulmonary edema. Authors could elaborate it or provide more insights within this article.

RESPONSE ==> We sincerely thank the reviewer for this valuable comment. Our previous study (Seo et al.) reported ciliary-related genes as candidates for ritodrine-induced side effects in a group that included patients with mild to severe cardiac and pulmonary dysfunction such as pulmonary edema, dyspnea, tachycardia, and palpitations. In this study, we narrowed our focus to patients with ritodrine-induced pulmonary edema (extreme cases) and identified two particular candidate genes associated with this side effect using matched controls. Contrary to the previous result, the two identified genes are not classified as ciliated-related genes and are still expected to be associated with ritodrine-induced adverse reactions, especially pulmonary edema. In the discussion, we attempted to explain how CPT2 and ADRA1A may contribute to the pathogenesis of pulmonary edema, but further research is needed to evaluate how these genes actually affect the action of ritodrine. We have added the limitation of the present study to the discussion.

[Correction in the revised manuscript] 

(In the Discussion, 6th paragraph)

… the current study. Future directions should focus on appropriate experimental evaluation of how candidate variants affect the pharmacokinetics of ritodrine and pulmonary edema. 

Q2. From the results the authors reveal that (CPT2 rs2229291, GG) , (CPT2 rs2229291, TG) and ADRA1A (rs2229126, TA) are significant to ritodrine-induced pulmonary edema. In discussion, authors try to interpret these clinical observations could be causative to the adverse effect in terms of basic studies (such as β2-knockout mouse and CPT2-deficient mouse models). How influential to these genetic variants it will be and any other previous cell-based studies mentioned the function of these genetic variants in different experimental settings.

RESPONSE ==> In previous studies, CPT2 rs2229291 has been known to exhibit reduced enzymatic activity under high fever. Yao et al.’s study showed that COS-7 cells transfected with a compound variant including rs2229291 exhibited decreased fatty acid b-oxidation, intracellular ATP levels, and mitochondrial membrane potential, which results in mitochondrial fatty acid metabolism disruption and brain edema in patients with influenza-associated encephalopathy. No previous cell-based studies were found for ADRA1A rs2229126. How these variants modulate variability in drug actions should be further evaluated in future studies. We have added the limitation of the present study to the discussion.

[Correction in the revised manuscript] 

(In the Discussion, 5th paragraph)

…in lung tissue. There has been a report of decreased activity of CPT2 with rs2229291 in hyperthermic conditions, resulting in impaired fatty acid oxidation and a mitochondrial energy crisis in vascular endothelial cells [34]; however, the direct functional impact of the variant on ritodrine-induced pulmonary edema should be further evaluated in future studies.

Q3. Certain statistics and linguistic could be fortified and precise.

RESPONSE ==> We thank the reviewer for this helpful comment. As suggested by the reviewer, a multivariate logistic regression was performed that adjusted for the cesarean delivery effect, where the p-value was less than .1 (p = 0.077) between groups (Supplementary Table S2). Even after adjustment, the p-value of CPT2 rs2229291 remained statistically significant (p = 0.048), whereas ADRA1A rs2229126 did not reach statistical significance. However, no formal adjustments were required, as there were no significant differences between groups in clinical features. We've also edited grammar, punctuation, and sentence structure to improve clarity. The comment is reflected on in the revised manuscript.

[Correction in the revised manuscript] 

(In the Results, Candidate variant prioritization and validation part)

had a marginal association signal (OR = Inf (1.0-Inf), Fisher’s exact test p-value = 0.053)… CPT2 rs2229291 remained statistically significant even after adjustment for the cesarean delivery effect (multiple linear regression p-value = 0.048, S2 Table).

---

## [Editor Report · Decision Letter 1]

12 Oct 2020

Identifying genetic variants associated with ritodrine-induced pulmonary edema

PONE-D-20-22975R1

Dear Dr. Kim,

We’re pleased to inform you that your manuscript has been judged scientifically suitable for publication and will be formally accepted for publication once it meets all outstanding technical requirements.

Kind regards,

Frank T. Spradley

Academic Editor

PLOS ONE

---

## [Editor Report · Acceptance letter]

29 Oct 2020

PONE-D-20-22975R1 

Identifying genetic variants associated with ritodrine-induced pulmonary edema

Dear Dr. Kim:

I'm pleased to inform you that your manuscript has been deemed suitable for publication in PLOS ONE. Congratulations! Your manuscript is now with our production department. 

Kind regards, 

on behalf of

Dr. Frank T. Spradley 

Academic Editor

PLOS ONE